# Design of a Compact and Minimalistic Intermediate Phase Shifting Feed Network for *Ka*-Band Electrical Beam Steering

**DOI:** 10.3390/s24041235

**Published:** 2024-02-15

**Authors:** Sebastian Verho, Jae-Young Chung

**Affiliations:** Department of Electrical and Information Engineering, Seoul National University of Science and Technology, Seoul 01811, Republic of Korea; sebastian@seoultech.ac.kr

**Keywords:** antenna array, beam steering, reconfigurable feed network, phase shifter

## Abstract

Intermediate phase shifting is a footprint- and cost-reduction technique for reconfigurable feed networks. These feed networks are utilized in antenna arrays to perform electrical beam steering. In intermediate phase shifting, a phase shifter is shared between two adjacent antennas. Conventionally, antennas only have individual phase shifters. With shared phase shifters, we reduce the number of components and the footprint by 25%. Consequently, this decreases the price and enables designs at millimeter-wave frequencies where space is limited due to frequency-dependent antenna spacing. This intermediate phase shifting is demonstrated by designing a reconfigurable feed network for the *Ka*-band that generates a continuous phase shift profile for beam steering. Due to the use of varactors and a novel biasing method, it does not require expensive beamformer integrated chips or lumped components for biasing. The feed network is combined with a 4 × 4 antenna array to demonstrate its beam-steering capabilities. The result is a high-density and minimalistic design that fits in a small volume of 25.6 × 25.6 × 0.95 mm^3^. With this small antenna array, the main beam is steered at $\pm 40^{\circ }$ broadside, providing full 1D and restricted 2D steering. It is a potential candidate for wireless sensor and mobile networks.

## 1. Introduction

The transition to millimeter-wave (mmWave) frequencies has enabled compact and small antenna array designs. Antenna arrays provide high directivity, which enables long-distance communications even with high atmospheric attenuation at high frequencies [1,2].

Reconfigurable antennas arrays provide more operational flexibility in the form of electrical beam steering or beam shaping, for example [3]. With electrical beam steering, we control the direction of radiated power and target it to the receiving end, improving reception. Although beam steering is often performed with antenna arrays, there have been many interesting alternate approaches recently, such as the Lunenburg lens [4,5], surface plasmons [6], plasmonic materials [7], leaky wave antenna [8,9], dielectric lenses [10] and ferroelectric materials [11].

In the *Ka*-band, research on antenna arrays with electrical beam steering is mostly focused on transmitarrays, reflectarrays and phased arrays [12,13,14,15,16,17,18,19,20]. Recently, reflective intelligent surfaces have also emerged as a candidate [21]. Nevertheless, transmitarrays and reflectarrays can be considered lenses that focus the radiated energy from a feeding horn antenna. In phased arrays, feeding is commonly performed with several cascaded power dividers.

Transmitarrays have been shown to have high aperture efficiencies of over 30% in mmWave frequencies [22]. However, they and reflectarrays have an inherently high profile due to the feed horn antenna. To address this, substrate-integrated waveguide (SIW) and integrated leaky-wave feed based solutions have been investigated [16,23]. However, they have a long focal distance (over 3$\lambda_{0}$) and thickness of 0.17$\lambda_{0}$, respectively. These are still high-profile. In addition, both feeds are waveguides that prevent the mounting of components. Therefore, they cannot be made reconfigurable and the reconfigurability must be implemented on another layer. This increases the profile, fabrication costs and design complexity.

Phased arrays have feed networks that are low-profile, and they allow the mounting of components, enabling reconfigurability. However, phased arrays use expensive beamformer integrated circuits (IC) to achieve this [13,24]. Transmitarrays and reflectarrays have addressed this issue with the use of simpler and cheaper components, e.g., p-i-n diodes and varactors [14,16,20,25]. There has also been research about the use of liquid crystal to scale designs to even higher frequencies [26]. The tradeoff of using p-i-n diodes and varactors, compared to beamformer ICs, is their reduced functionality and the biasing network they require, which has a greater footprint. Beam steering is also possible without active components with a butler matrix [27,28], for example.

Considering the above discussion, we develop a reconfigurable *N*-by-*N* feed network for the *Ka*-band. It utilizes the low-cost components found in transmitarrays while also being low-profile like the feed networks of phased arrays. This presents a design challenge of the *Ka*-band since the available space is dictated by the frequency-dependent antenna spacing [24]. For example, a 4 × 4 antenna array with a 0.5$\lambda_{0}$ antenna spacing has a total footprint of 21.4 × 21.4 ${\mathrm{mm}}^{2}$ available for the feed network design at 28 GHz. This small footprint must contain the power divider network along with several phase shifters and their biasing network to be a low-cost reconfigurable feed network. Therefore, a high-density design is paramount. Other similar works [29,30] operating in the *Ka*-band were not able to achieve this. They fabricated and measured only passive versions of the original reconfigurable designs. The reason is likely the lack of space for implementation.

To achieve a reconfigurable feed network in the *Ka*-band, a compact design is required. Therefore, we propose a phase shifting scheme coined intermediate phase shifting. Its basic idea is to share a phase shifter between two adjacent antenna elements. This reduces the footprint and the number of components by 25% compared to using a two-stage phase shifter for each antenna. When utilized in a small antenna array, it provides phase shifting for full 1D beam steering and restricted 2D beam steering. Additionally, we employ a novel biasing method which does not require any lumped components, reducing the footprint usage even further. This results in a high-density and minimalistic reconfigurable feed network. Its footprint is less than 2.4 × 2.4$\lambda_{0}^{2}$ (25.6 × 25.6 mm^2^) with a thickness of 0.028$\lambda_{0}$ (0.3 mm) at 28 GHz. This design offers a compromise between small size, simplicity and performance.

To demonstrate the operation of the intermediate phase shifter feed network, we combine it with a simple 4 × 4 antenna array to form a prototype. We present far-field radiation pattern beam-steering results at several steering angles for the full structure. This work is the first to successfully utilize varactors in the *Ka*-band to the best of the authors’ knowledge. In [25], varactors were used in a *K*-band design, but there was no gain in the far-field measurements.

This article is organized as follows. First, we present the intermediate phase shifting concept used for beam steering and footprint reduction in Section 2. This leads to a discussion about the full 4 × 4 antenna array structure in Section 3. This section contains details about the proposed feed network structure and provides a discussion about choosing the right varactor model for *Ka*-band designs and other design considerations. In Section 4, we present the beam-steering strategy and measurement results of a separate phase shifter circuit. This is utilized in determining the phase shift profile used in the far-field measurements of Section 5. Lastly, we provide further discussion about the design and a comparison to other related works in Section 6.

## 2. Intermediate Phase Shifting

Intermediate phase shifting is implementable with any kind of phase shifter that is integrated with a feed network. In this work, we focus on a varactor-based phase shifter as it provides a continuous phase profile, increasing design compactness. The varactor is a voltage-tunable capacitance. However, its capacitance range is restricted, meaning it provides a limited, finite phase shift. This is due to the depletion region of the varactor having a physical limit on the extent to which its width can be altered.

There has been plenty of research aiming to extend the total phase shift of varactor-based phase shifters [31,32,33,34,35,36]. However, achieving the required 360° of phase shift always requires an increase in the number of varactors either by having additional phase shifter stages or adding more varactors to a single stage [33]. This increases the overall cost and footprint of a design.

To address the above, we propose a concept coined intermediate phase shifting. The basic idea of it is simple, and the concept is illustrated in Figure 1. A typical approach is shown in Figure 1a, where the input signal is split into two branches by a power divider. Both branches have a two-stage phase shifter which can be connected to an antenna, for example. Since the two branches share a common conduction path through the power divider, we can move one phase shifting stage from each branch to in front of the power divider input and combine them, as illustrated in Figure 1b. This way, the overall number of phase shifters is reduced by one, i.e., 25%. Consequently, the number of varactors is also reduced by 25%. Fewer phase shifters means a smaller footprint, as well.

The above result is significant since other *Ka*-band reconfigurable feed networks [29,30] are missing complete biasing structures and components in their fabricated designs. The potential reasons are the lack of space or difficulties in obtaining the biasing. We address both issues in this work. Nevertheless, Figure 1c represents the final intermediate phase shifting concept. Henceforth, the phase shifters before and after the power divider will be called the intermediate phase shifter and independent phase shifter, respectively.

Having a shared phase shifter (i.e., intermediate phase shifter) between two branches creates a dependency between them. This is because controlling the intermediate phase shifter alters the phase shift of both branches. Therefore, independent phase shift control for each branch is only possible with the adjustment of the non-shared phase shifter, i.e., the independent phase shifter. This restricts the beam steering as only a limited amount of phase shift can be generated between the branches. More specifically, if the phase shift is <360° for the independent phase shifter, it is not possible to achieve the full $360^{{}^{\circ}}$ phase range required for unrestricted beam steering. The severity of this limitation depends on the phase shift range of the independent phase shifter. This will be demonstrated later in this section.

In Figure 1d, the feed network is expanded for the control of a 2 × 2 antenna array. Consequently, it has branches with independent and restricted beam steering in the horizontal and vertical directions, respectively. In the latter, the phase shift generation is still restricted due to the phase dependency from the shared intermediate phase shifter, as discussed above. However, the horizontal direction does not have this limitation. This is because the control of the intermediate phase shifter on the left side does not affect the phase shift of the right-side branches in the horizontal steering direction, i.e., independent phase tuning between branches. Therefore, it is possible to perform unrestricted 1D and restricted 2D beam steering.

To provide insight into the steering limitation in the vertical direction, the result is presented in Table 1. It shows the relation between maximum progressive phase shift $\psi_{\max}$ and maximum steering angle $\theta_{\max}$. This was evaluated for the 2 × 2 array with
(1)$$|\theta_{\max}|=\left|\arcsin\left(\frac{\psi_{\max}}{\pi }\right)\right|.$$

This is derived from (Equation 6), introduced later in Section 4, using common $0.5\lambda_{0}$ array spacing. Three maximum progressive phase shifts $\psi_{\max}$ of 30°, 90° and 180° are evaluated with (Equation 1). This, respectively, leads to three maximum steering angles: ($\theta_{\max}$) $\pm 9.6^{\circ }$, $\pm 30.0^{\circ }$ and $\pm 90.0^{\circ }$. From this result, we observe the importance of generating adequate phase shift to have a wide beam-steering range. However, this analysis only considers the phase and does not contain the limitations imposed by other factors such as grating lobes or scan loss. Nevertheless, this result demonstrates the beam-steering limitation of intermediate phase shifting in the vertical direction.

## 3. The 4 × 4 Antenna Array Prototype

Characterizing only the 4 × 4 intermediate phase shifter network would be difficult due to its small size. Therefore, a 4 × 4 antenna array prototype was designed to operate at 28 GHz to demonstrate the feasibility of the intermediate phase shifting feed network through beam steering. An initial simulation study of a similar structure was conducted in [37].

The complete structure containing the feed network and passive patch antennas is shown in Figure 2. It consists of four layers designed on High-Temperature Co-fired Ceramic (HTCC), whose electrical properties are $\varepsilon_{r}=8.5$ and tanδ=0.001. Unlike in printed circuit boards (PCB), the conductor is made of tungsten ($\sigma =18.2\times 10^{6}$ S/m). The structure is fed with a SubMiniature version A (SMA) connector.

HTCC was chosen due to its high permittivity and low dielectric loss. Additionally, it has the benefit of having more freedom in controlling layer thicknesses. In contrast, the more common PCBs are available only in a limited number of thicknesses. With HTCC, layers as thin as 0.1 mm are possible. Thin layers are beneficial since they increase the cut-off frequency of surface waves, reducing the number of spurious modes.

The thin layers and the high permittivity provide narrow and short transmission lines (tx-line), increasing the design density. Therefore, expensive fine-detail fabrication techniques are not required, e.g., complementary metal–oxide–semiconductor (CMOS) and monolithic microwave integrated circuit (MMIC). As mentioned earlier, high density is a key factor since the footprint of an antenna array is determined by the frequency-dependent array spacing, i.e., there is less physical space for a design at higher frequencies.

### 3.1. Varactor Selection

Typically, there is not much discussion about component selection for beam-steering applications employing varactors and p-i-n diodes. For example, *Ka*-band designs commonly use the p-i-n diodes MADP-000907 and MA4AGP907 but do not provide a reason for their selection [14,15,16,20]. Furthermore, varactors have not been utilized in beam steering in the *Ka*-band to the best of the authors’ knowledge.

Selecting the right component is an important part of the design as it enables the design of reconfigurable structures. Moreover, these components contain parasitics which increase losses, shift the operating frequency, and affect the phase [38]. Therefore, we discuss some critical considerations regarding the effect of the parasitic-inductance to varactor phase shift. Not considering the inductance in mmWave frequencies can lead to a failed design. Hence, understanding it is critical.

To analyze the effect of the parasitic-inductance to varactor phase shift, we show their relation through a discussion based on analytical equations below. We start the analysis by looking at the general equation for calculating the phase angle from impedance. It is
(2)$$\theta =\arctan\left(\frac{X}{Z_{0}}\right)$$
where *X* is the reactance and $Z_{0}$ is the system impedance. For varactors, the reactance *X* can be simply expressed with the reactive inductance $X_{L}$ and the reactive capacitance $X_{C}$ as follows
(3)$$X\left(V\right)=X_{L}-X_{C}=2\pi fL-\frac{1}{2\pi fC(V)},$$
where *V* is the bias voltage, *f* is the operating frequency, *L* is the inductance, and *C* is the capacitance [39]. As is evident from (Equation 3), the reactive inductance $X_{L}$ becomes dominant over the reactive capacitance as frequency increases. If $X_{L}>>X_{C}$, the total reactance becomes insensitive to changes in capacitance, i.e., there is no phase shift. Reducing inductance is an obvious solution to increase phase shift, but often, it is impossible to avoid certain inductive components, e.g., package inductance, solder inductance or inductance from traces and vias. Therefore, the reactive inductance $X_{L}$ can be counteracted only by choosing a varactor with a low tunable capacitance.

To illustrate the effect of increasing inductance, an analytical comparison based on (Equation 2), (Equation 3) and the varactor equation [34] is presented in Figure 3a for 28 GHz. It contains the phase profiles of two varactor models in a single-stage phase shifter configuration. Their capacitances are 1.1–0.17 pF (Varactor 1) and 0.2–0.03 pF (Varactor 2), respectively, with a voltage bias of 0–12 V. The capacitance values of the varactors are closely related to varactor models MAVR-000120-1411 and MAVR-011020-1411, respectively. The other varactor parameters are assumed to be identical for a fair comparison and for the sake of this example.

To understand the effect of inductance on phase shift, two inductance values are assigned to each varactor model and compared to each other. The values 0.2 nH and 0.4 nH are chosen. The former represents a package inductance value [40]. The latter represents the package inductance combined with some additional parasitic inductance such as solder. This value was also used in [41], for example.

The phase shift is shown in Figure 3a for the two cases. As can be seen, the increase in inductance decreases the phase shift of Varactor 1. However, the opposite effect is observed for Varactor 2 due to the reactive inductance becoming more similar in magnitude to the reactive capacitance, i.e., higher phase shift. Hence, it is critical to consider the relation and magnitude of reactive elements to each other when choosing a varactor for a phase shifter. Otherwise, there is a risk of not acquiring enough phase shift for beam steering, which would limit the scan range. In our example, Varactor 2 would still be able to fulfill a phase shift requirement of 180° even if the inductance was incorrectly estimated. However, this would not be the case for Varactor 1, which has too little phase shift to be used effectively in beam steering.

The inductance also affects the reflection coefficient of the varactor (i.e., losses). Varactors are commonly utilized in reflection-based phase shifters. The varactor reflection coefficient $\Gamma_{\mathrm{varactor}}$ is described by the equation
(4)$$\Gamma_{\mathrm{varactor}}=\frac{Z_{\mathrm{varactor}}-Z_{0}}{Z_{\mathrm{varactor}}+Z_{0}}.$$

The $Z_{\mathrm{varactor}}$ is the total impedance of the varactor which contains the reactance *X* and a resistive component *R*. Plotting the two cases of Varactor 1 in Figure 3b shows the varying reflection coefficient when R=8Ω. The voltage dependency of the reflection coefficient is evident. When $X_{L}=X_{C}$, the reflection coefficient is at its lowest value (i.e., highest loss). For the L=0.2 nH case, this is around 5.5 V. Increasing the inductance value to 0.4 nH shifts this highest loss point to around 12 V. If the inductance can be controlled, it presents two design options. First is the 0.2 nH case. It has less loss variation in the whole voltage range. However, the average loss is higher than in the 0.4 nH case. Lower loss unfortunately also leads to a smaller phase shift, as observed in Figure 3a. Therefore, a compromise between losses and total phase shift is required.

As is evident from (Equation 3), the total phase shift is also frequency-dependent. This has been illustrated in Figure 3c for both varactor models with the two inductance values. The frequency range is from 20 GHz to 36 GHz. At 28 GHz, Varactor 1 with 0.4 nH inductance has the lowest total phase shift. Due to the relation between reactive inductance and reactive capacitance, it surpasses the two other cases for frequencies below 23 GHz. For Varactor 2, the inductance of 0.4 nH increases the total phase shift for the whole frequency range compared to 0.2 nH. From these observations, we conclude that understanding the frequency-dependent behavior is critical for optimal wide-frequency-band applications.

Based on the above phase shift analysis, we searched for suitable varactor models. The objective was to find a model that has low enough capacitance to counteract the package and other parasitic inductances in the *Ka*-band. This guarantees adequate phase shift generation for beam steering. However, most varactors available on the market have too high capacitance for *Ka*-band phase shifters. In the end, only the varactor MAVR-011020-1411 was deemed to have a low enough capacitance. Its capacitance range is from 0.19 pF to 0.025 pF (0–15 V) [42]. By minimizing the inductance, it might be possible to use other models as well. However, there is a risk of having higher-than-expected inductance based on the authors’ previous experiences. Therefore, the model discussed above was chosen and its operation is validated later in Section 4.2.

### 3.2. Varactor Biasing

Bias-T structures are commonly used for biasing varactors. They often consist of a combination of lumped resistor–capacitor–inductor (RLC) components. These are problematic in the *Ka*-band and higher frequencies due to their package parasitics, which degrade their performance or make them unusable. These parasitics can be reduced by making smaller packages, but the size reduction is limited by fabrication technology. Additionally, even the small 0201 component package (0.6 mm × 0.3 mm) is approximately 0.056$\lambda_{0}$ long at 28 GHz. This is 11.2% of the typical 0.5$\lambda_{0}$ antenna array spacing, occupying significant footprint.

Considering the above, we investigate options requiring no RLC components and which are directly integrated into the structure. Common solutions are radial stubs, meander lines and distributed capacitors [16,20,43]. While these are valid approaches, the design of this paper requires a way to control each phase shifter individually. Implementing this is not an issue in transmitarrays, for example, since the antennas are not connected to each other with a common conduction path. However, in this work, each feed network branch feeding an antenna is in the same direct current (DC) potential due to the shared conduction path. In previous work [44], this was resolved by adding lumped capacitors between the branches, but, as mentioned earlier, the objective is to avoid them. Therefore, an alternative solution composed of an interdigital band-pass filter and butterfly radial stubs is proposed and described below.

Varactor biasing requires a ground connection. Inductors are commonly added to this connection to prevent interference with the high-frequency radio frequency (RF) signal. Instead, we replaced them with the interdigital band-pass filter presented in [45] and illustrated in Figure 4a. This is a novel way to use this component, and the benefit is the elimination of the inductor. Additionally, it can filter out-of-band interference.

The interdigital band-pass filter consists of parallel 90° electrically long tx-lines. The lines are open and short-circuited on opposite ends. The short-circuit is made with a via. The structure is also compact in size [46].

The design of the structure started by setting the $w_{\mathrm{gap}}$ in Figure 4a to the minimum gap width of 0.1 mm. This ensures minimal insertion loss by tightly coupling the parallel lines, i.e., fewer fringing electro-magnetic (EM) fields. Additionally, the diameter of the two vias was also set to the smallest possible value of 0.1 mm to decrease the overall footprint. Lastly, $l_{\mathrm{line}}$ and $w_{\mathrm{line}}$ shown in Figure 4a were optimized with simulations in Ansys HFSS.

The simulation results are presented in Figure 4b. The $|S_{11}|$ and $|S_{21}|$ are −16.1 dB and −0.6 dB at 28 GHz, respectively. The $|S_{11}|$ stays below −10 dB from 24.2 GHz to 32.4 GHz. This is a fractional bandwidth of 29.3%. As seen from the results, this is a suitable replacement for the via-terminated inductor as it is low-loss and has a wide operational bandwidth.

For the positive terminal of the varactor biasing, radial stubs were employed. A butterfly configuration was chosen to enhance the isolation of the RF signal. This was found to be most space-efficient through simulations. Its footprint was minimized by rotating the stubs closer to each other. This increases capacitive coupling and consequently lowers the operating frequency. Therefore, the stub size can be reduced without compromising performance. The optimized design is shown in Figure 5a which is placed under the phase shifter in L2 shown in Figure 5b. Via V1 connects the bias line to the phase shifter. Via V2 represents the ground connection provided by the interdigital band-pass filter.

For separating the positive terminal and ground from each other, the internal capacitance of the varactor is utilized, as illustrated with the equivalent circuit in Figure 5c. The varactor capacitance is an open circuit for DC signals and ensures the ability to create a voltage potential difference between the terminals. The antenna and the gap in the interdigital band-pass filter also act as capacitors (i.e., open connections) for the DC signals, providing isolation.

Other recent works utilizing feed networks [47,48,49,50] have used RLC components in biasing. Therefore, the designs were limited to low frequency bands below 10 GHz. As discussed before, the similar *Ka*-band designs [29,30] did not contain a full design. Our biasing method provides a way to implement fully fabricatable and functional designs in the *Ka*-band.

### 3.3. Reconfigurable Feed Network

The reconfigurable feed network shown in Figure 6 consists of two base designs—phase shifter and power divider. There is also the interdigital band-pass filter discussed in the previous subsection.

The power dividers are cascaded in an H-tree configuration to divide the input power equally between the 16 antennas, i.e., −10log16=−12.0 dB at each output in an ideal lossless case. The T-junction power divider was chosen due to its small footprint compared to the Wilkinson power divider. Additionally, this eliminates the need for a resistor, which simplifies the design even further. The drawback is the increased coupling between antennas, which is expected to degrade the beam-steering performance by limiting the steering angle.

A single power divider was designed to have 50 Ω input and outputs which are equal to a line width of 0.34 mm with a substrate thickness of 0.3 mm. Consequently, the 35 Ω quarter-wave transformer had a length and width of $l_{\mathrm{trans}}=$ 1.0 mm and $w_{\mathrm{trans}}=$ 0.7 mm, respectively. In simulations with Ansys HFSS, the insertion loss and reflection loss were 3.2 dB and 40 dB at 28 GHz after optimization.

Varactor-based phase shifter designs often use a 90° branch line coupler. It is a robust design which provides a wide phase shift range and low insertion loss. However, it occupies a large footprint due to the four quarter-wavelength tx-lines, and this approach also limits its bandwidth [51,52]. A miniaturized version of it was presented in [53], but an even smaller design is required. Therefore, we utilized a phase shifter design based on two coupled lines and three varactors from [51]. It has been used in [54] for beam steering. A modified version of the phase shifter is shown in Figure 6. Typically, phase shifter vias are terminated to the RF ground. However, to implement the discussed biasing method, the vias were connected to the radial stubs.

The design originally had layers L2 and L3 of Figure 5b in an opposite arrangement to increase isolation between the tx-lines and the bias lines. However, this was observed to generate strong surface waves due to the RF signal coupling with the ground, i.e., the via penetrating the ground acted as an aperture coupler with the via clearance. Hence, the current configuration of Figure 5b was chosen.

Ideally, the two coupled lines of the phase shifter would be 90° long electrically (≈1.1 mm) at the design frequency of 28 GHz. However, a shorter design was created with optimization due to the footprint constraints. The objective was to obtain greater than 10 dB return loss and over 180° of phase shift. This also guarantees low insertion loss. In other words, there is a trade-off between phase shift and insertion loss. More phase shift can be obtained with higher insertion loss.

The phase shifter optimization process was performed with Ansys HFSS. It started by fixing the $w_{\mathrm{gap}}$ to 0.2 mm to ensure tight coupling and to ease the soldering of the two parallel varactors. The phase shifter parameters $l_{\mathrm{line}}$ and $w_{\mathrm{line}}$ were optimized to 0.8 mm and 0.1 mm, respectively. The insertion loss ranged from 1.1 dB to 3.1 dB in the simulations when the phase shifter was biased from 0 V to 15 V. The return loss was above 20 dB for all voltages. The optimized structure was also confirmed to provide over 180° of phase shift.

The power dividers, phase shifters and interdigital band-pass filters were added on a single layer to form the feed network. On the layer between the feed network and ground, the bias lines were added. To ensure minimal interference with the tx-lines, the bias lines were placed close to the RF ground at a distance of 0.1 mm, as shown in Figure 2.

The allotted footprint for the feed network was determined by antenna spacing. Initially, it was set at 0.5$\lambda_{0}$ (at 28 GHz). This is enough space for the feed network. Next, the bias lines for the phase shifters were carefully routed by avoiding any vertical overlaps with the tx-lines. This reduced coupling between them and decreased the effect of the bias lines on the tx-lines. However, the horizontal distance between them was not enough around the input feed. Therefore, the coupling between them was strong enough to alter the characteristic impedance of the tx-lines. Moreover, some of the input power leaked to the bias lines. Consequently, the return loss degraded to less than 10 dB. This issue was avoided by setting the antenna spacing to approximately 0.6$\lambda_{0}$ (at 28 GHz). The final design is shown in Figure 6, where the bias lines are colored in gray.

The full feed network of Figure 6 was simulated with Ansys HFSS. The results are shown in Figure 7. The average insertion loss for the 16 outputs at 28 GHz was 7.0 dB at 0 V bias. The loss is mostly from the phase shifters and the conductor. Both have a loss of 2.2 dB, as shown in Table 2. The conduction loss was determined by comparing the simulation results. First, the conductors were set as perfect electrical conductors. This result was compared to pure tungsten (i.e., the conductor material). With this, the conduction loss of pure tungsten was found to be 2.2 dB.

Lastly, we summed up the losses of the power divider, the interdigital band-pass filter, the phase shifter, and the conductor from separate simulations. The total loss was 5.8 dB. Therefore, the remaining 1.2 dB of loss is believed to be from the unaccounted for radiation and surface wave loss of the feed network. Back-radiation is a known issue with microstrip lines. It is caused by the loosely coupled fields of the structure. Similar amounts of radiation and surface wave loss were observed in [55] for a 16-element antenna array feed network, agreeing with our presumption.

The simulated reflection loss was 23.7 dB with a 10 dB bandwidth of 0.5 GHz equal to a 1.8% fractional bandwidth, as shown in Figure 7a.

Next, we studied the effect of phase shifter voltage control on the feed network losses. For this, the voltage of each phase shifter was simultaneously altered from 0 V to 15 V in 1 V increments. It should be noted that these do not represent different beam-steering angles since the phase of each phase shifter is identical, i.e., 0${\;}^{{}^{\circ}}$ steering angle. As seen in Figure 7b, 0 V has the lowest average insertion loss of 7.0 dB (12 dB from power division not included). The loss has an increasing trend with an increase in voltage. This is due to the lower reflection of the varactor, as discussed in Section 3.1. The highest loss is 13.4 dB at 13 V. The insertion loss does not monotonically increase due to the varying return loss. This variation is caused by the resonances in the feed network structure. The return loss varies from 7.5 dB to 22.1 dB.

The insertion loss is an important metric in determining the performance of the feed network. In [25], the losses of a varactor-based unit cell of a transmitarray varied approximately from 5 to over 10 dB in simulations. According to the measurements, this loss was between 10 and 20 dB. The losses in [29] were similar to our design, with a p-i-n diode-based design in the simulations and measurements, respectively. This is an interesting result as the measured results did not contain the losses of the p-i-n diode since the components were not added. Based on this comparison, it can be concluded that the presented feed network in this work can be utilized in *Ka*-band applications. It provides a compromise between loss and size.

## 4. Beam Steering and Phase Shift Characterization

### 4.1. Beam Steering

The general 2D beam-steering equations for determining progressive phase shift ψ between unit cells with continuous phase control are
(5)$$\left\{\begin{array}{c} \psi_{x}=-k_{0}p\sin\left(\theta \right)\cos\left(\phi \right)\\ \psi_{y}=-k_{0}p\sin\left(\theta \right)\sin\left(\phi \right). \end{array}\right.$$

$k_{0}$ and *p* are the wave number and unit cell spacing, respectively. θ and ϕ are the steering angles of interest in the spherical coordinates [56]. For 1D steering in the E-plane ($\phi =90^{{}^{\circ}}$), (Equation 5) reduces to
(6)$$\psi_{x}=-k_{0}p\sin\left(\theta \right).$$

For a typical design not utilizing intermediate beam steering, an integer multiple *N* (=0, 1, 2, *…*) of (Equation 6) determines the required phase shift generated by a two-stage phase shifter. This is illustrated in Figure 8a with an example 2 × 4 array. Each stage generates half of the required phase shift. For the intermediate phase shifting, the control is similar, as shown in Figure 8b. The intermediate phase shifter generates half of the required phase shift for two branches instead of having one additional stage in both branches.

The above is one way to control the phase shifters. It is also possible to generate unequal phases with the independent and intermediate phase shifters as long as the total phase shift is the one calculated with (Equation 6). However, this complicates the control as it requires calculating two control voltages instead of one. Additionally, this creates unequal insertion loss between the two phase shifters based on the result in Figure 7b. Consequently, the average loss increases since the relation between loss and bias voltage is not linear. Therefore, the control presented in Figure 8b is utilized later in Section 5.

### 4.2. Phase Shift Characterization

To generate the progressive phase shift calculated with (Equation 6), the phase versus bias voltage of the phase shifter must be known. Moreover, the phase shifter is the only reconfigurable component in the feed network. Therefore, it is important to understand its effect when it is later used in beam steering. For this, a board containing two phase shifters was fabricated. It is shown in Figure 9a and it has the same biasing scheme as that introduced in Section 3. A two-port measurement was performed with an Anritsu MS46122b vector network analyzer (VNA).

The $|S_{11}|$ and $|S_{21}|$ results are shown in Figure 10 for simulation and measurement when one phase shifter was controlled from 0 V to 12 V. For determining the bandwidth of the phase shifter, we used two criteria. The first one is less than 10 dB of insertion loss. The second one is a maximum 2 dB difference in insertion loss between the bias voltage states at a single frequency point. A small insertion loss difference improves the radiation pattern of an array during beam steering. The gain increases and side lobe levels (SLL) are reduced due to constructive interference of the radiated waves from each antenna. For the simulation, this 2 dB flatness criterion equals a bandwidth of 2.9 GHz from 26.9 GHz to 29.8 GHz, as shown in Figure 10b. The $|S_{21}|$ varies from −7.2 dB to −3.3 dB in this frequency range. This simulation contains losses from the two SMA connectors and interdigital band-pass filter, as well.

In the measurement, the frequency bandwidth was narrower than in the simulation, as is evident in Figure 10d. With the simulations, similar behavior was obtained by increasing the inductance of the varactor model from 0.5 nH to 1.2 nH. This increase is likely from the unaccounted for inductance of the solder. It degrades the matching of the phase shifter since it is not optimized for high inductance. Therefore, the $|S_{11}|$ is above −10 dB in Figure 10c for 9 V and 12 V bias voltages.

Based on the 2 dB criteria, the frequency band is from 24.7 GHz to 26.9 GHz, i.e., the bandwidth is reduced from 2.9 GHz to 2.2 GHz compared to simulations. The $|S_{21}|$ varies from −9.2 dB to −5.5 dB in this range. The increased loss compared to the simulations is believed to be from higher-than-expected varactor losses, higher conductor resistivity and unaccounted for surface roughness. Additionally, bias voltages at 6 V and above also have lower reflection losses due to the high inductance discussed above.

Due to the phase shifter bandwidth reduction, the phase shift was characterized at 26.7 GHz, and this frequency is later used in Section 5 for beam steering. The characterization was performed the a curve fit method, presented in previous work [44]. This determines the varactor parameters and provides a way to calculate the phase shift versus voltage profile analytically. The obtained varactor parameters are presented in Table 3. With them, we obtain the curve fit result which is shown in Figure 9b along the phase shift measured at 26.7 GHz. They are in good agreement and the mean absolute error is 0.9°.

Biasing the phase shifter from 0 V to 15 V produces around 190° of phase shift. We obtain the required 180° of phase shift around 12 V, which is the maximum control voltage in beam steering. Based on the measurement results of Figure 10c,d, control voltages of 9 V and above significantly degrade the gain and SLL of the antenna array. Therefore, their use is avoided when possible by calculating the phase shift in relation to 0 V. Lastly, it should be noted that a single antenna will be controlled by two phase shifters, i.e., a 360° total phase shift.

The phase shift is shown for two other frequencies, 25 GHz and 28 GHz, as well. As discussed in Section 3.1, the phase shift is dependent on the relation of the inductive and capacitive components. Both are frequency-dependent, as was predicted with the analytical analysis in Figure 3c and confirmed with the measured phase shift in Figure 9b. At the lower frequency of 25 GHz in Figure 9b, the magnitude of the inductive reactance $X_{L}$ decreases, which consequently increases the phase shift. The opposite is true for the 28 GHz phase shift curve.

## 5. Chamber Measurements

The fabricated 4 × 4 antenna array prototype is shown in Figure 11. Four pin headers were mounted to enable interfacing with two EVAL-AD5766SD2Z digital-to-analog converter (DAC) boards. They provide 0–20 V reverse voltage to individually control the 24 phase shifters on the feed network. The DAC boards are controlled with a laptop, which communicates with the universal serial bus (USB) protocol. A MATLAB script was created to automate the calculation of the required control voltages for beam steering. It utilizes the phase shift characterization of the previous section.

The far-field measurement setup is shown in Figure 12. The 4 × 4 antenna array was mounted on a platform that rotates from −90° to +90° in the horizontal direction. The resolution of the measurement angle was set to 1°.

The 4 × 4 antenna array gain versus frequency was measured at 0° broadside by biasing all the phase shifters, first at 0 V, and then, at 15 V, i.e., zero progressive phase shift. As was shown in the phase shifter measurement of Section 4.2, higher bias voltages have higher insertion loss. The 0 V biasing demonstrates stable operation in the whole frequency band from 26.5 GHz to 28.0 GHz, where the gain fluctuates between 6.3 dB and 8.0 dB, as shown in Figure 13a. For the 15 V biasing, there is larger variation between the maximum and minimum gain, which are 6.9 dB and −4.1 dB, respectively. This is due to the decreased bandwidth of the phase shifter, as discussed in Section 4. At 27.0 GHz, the gain drops 0.4 dB from 7.3 dB between the two biasing cases. Therefore, the loss is low at the beam-steering frequency of 26.7 GHz. Moreover, 15 V is not used in the beam steering, as we obtain 180° of phase shift at 12 V already. The maximum voltage used in beam steering is 8.6 V which is used in the $\pm 50^{\circ }$ steering angles discussed later.

The co- and cross-polarization levels for the 0° steering angle were measured, as well. As seen in Figure 14a, the difference between the levels is over 22 dB at 0°. The −3dB bandwidth is approximately 25° and the cross-polarization discrimination level is above 15 dB in this range.

The beam steering was successfully performed from −40° to +40° in the E-plane ($\phi =90^{{}^{\circ}}$) direction with a peak gain and scan loss of 7.8 dB and 4.7 dB, respectively. The results are shown in Table 4, where the scanning angle followed the control angle well from −20° to +20°. Steeper angles required a larger-than-calculated steering angle. For example, an angle of 40° required 50° of steering from the controller side. This discrepancy results from the mutual coupling between antennas. A Similar observation was made in [30]. The feed network does not have Wilkinson power dividers due to the lack of space. Therefore, the coupling between the antenna elements increases, which limits the beam steering and increases the SLLs [57,58]. The SLL increase is especially notable for steering angles beyond −30° and +10°. The mutual coupling negatively affects the progressive phase shift between the antenna elements. This effectively prevents the deconstructive interference of radiated waves from canceling the side-lobes.

The side-lobes can be further suppressed by adding tapering to the feed network and by increasing the number of antenna array elements.

For steering angles of $\pm 40^{\circ }$, $\pm 20^{\circ }$ and 0°, a comparison with the simulations is shown in Figure 14b. The results show good agreement with a gain difference of 0.7 dB for the 0° steering angle.

In Figure 13b, the fairly stable gain profile of the beam steering is shown. The steering angles of 0°, −20° and −40° are included. Interestingly, the −40° steering angle has a higher gain than the −20° steering angle at 28 GHz. This is due to the frequency- and voltage-dependent insertion loss of the phase shifter since each steering angle has a different set of control voltages, as shown in Table 4.

## 6. Discussion

The aperture size of the 4 × 4 antenna array is 25.6 × 25.6 mm${\;}^{2}$. Therefore, the ideal directivity at 26.7 GHz is 18.1 dBi. With a gain of 7.6 dBi in the 0° direction, the aperture efficiency is 8.8%. As was shown in Table 2, the main loss sources were phase shifter and conduction loss. The latter is common in feed network-based designs due to the signal propagating several wavelengths along the lossy conductor. In return, the profile of the structure is reduced significantly, as discussed later.

The most relevant works operating in the *K*- and *Ka*-bands with beam-steering functionality are presented in Table 5. Moreover, their main performance metrics are compared in Table 6. It should be noted that the most similar works in the *Ka*-band are [29,30]. However, they were not able to fabricate the full system since the reconfigurable components were replaced with short or open connections. Additionally, they are limited to a 1D array. Both utilize p-i-n diodes for beam steering. The downside of them compared to varactors is their higher DC power consumption and limited number of phase states. Without a continuous phase profile, there need to be more antenna elements to achieve a wide beam-steering range. Our design is able to provide an adequate scan range with a small antenna array size.

So far, [25] operating in the *K*-band is the highest frequency band where varactors have been utilized for beam steering. However, the authors want to point out that [25] was not able to obtain any gain in measurements. Therefore, to the best of the authors’ knowledge, the work presented here has the highest frequency band where varactors have been successfully utilized in beam steering.

The design in this work is also very thin, with a thickness of only 0.95 mm, which, at 26.7 GHz, equals 0.085$\lambda_{0}$. The thickness of the feed network is 0.3 mm (0.027$\lambda_{0}$). As is evident in Table 6, the design in this work has the smallest profile. Therefore, our 4 × 4 antenna array can be considered very thin, and it is suitable for applications requiring a low profile.

The beam-steering range of the proposed design is $\pm 40^{\circ }$. This is a good result for a small array size. For example, Ref. [16] has 25 times more antenna elements, but it has the same beam-steering range while having a high-profile structure.

The prototype array presented here is only 4 × 4 in size, but it can be expanded to larger array sizes by having the feed network as a part of a 2 × 2 or 4 × 4 subarray. A larger array would increase the gain and decrease the SLLs.

Comparing the aperture efficiency $\varepsilon_{\mathrm{ap}}$ of the different works shows wide variation in Table 6. The highest $\varepsilon_{\mathrm{ap}}$ is in [13]. However, it should be noted that the design contains amplifiers. This distorts the comparison as other works do not have amplifiers. Nevertheless, the work in this paper demonstrates higher aperture efficiency than [16,25]. In [29,30], the efficiency is higher, but, as discussed before, the designs lack full implementation. Therefore, it can be concluded that the work presented here offers relatively good performance.

Thanks to the varactor having controllable capacitance, it consumes a negligible amount of DC power. Therefore, it has been marked as having low power consumption in Table 5. p-i-n diode-based designs commonly use the model MA4AGP907. When driven with a 10 mA current, the power consumption is typically 13.3 mW, which we rate as medium power consumption [59]. The highest power consumption is described in [13] due the use of an beamformer IC which consumes 0.5 W or 0.8 W per component [60]. Power consumption generates heat and might require a heat sink to dissipate it. This increases the cost and the profile of the system. Therefore, it is beneficial to have low DC power consumption. The tradeoff is the lack of amplification in the varactors and p-i-n diodes.

This design only uses one component type thanks to its novel biasing arrangement providing a minimalistic design. Simple interfacing is also possible with the SMA connector at the input feed.

The varactor is a cheap component compared to beamformer ICs which are utilized in [13], for example. Although other works typically have higher aperture efficiency, the design in this work provides a compromise between efficiency and small size. The profile is over 95% smaller than designs that are based on transmitarrays. Therefore, the design in this work can be considered low-cost and small-size.

## 7. Conclusions

This work proposed a novel reconfigurable feed network for electrical beam-steering applications in the *Ka*-band. The main beam was approximately steered from −40° to +40° in the measurements. Even with the small physical size of 25.6 × 25.6 mm${\;}^{2}$, it has a relatively wide scanning range while being less than 1 mm thick. Other works have attempted to create a compact design for the *Ka*-band before, but were unsuccessful, since there is little footprint available due to the frequency-dependent array spacing. The biasing technique and intermediate phase shifting presented in this paper provide a solution to achieve a compact antenna array. The design is also minimalistic as the varactor is the only component type. This is also the only work so far to utilize varactors in the *Ka*-band for beam steering. Varactors provide a continuous phase profile and consume almost no DC power, being superior to the more common p-i-n diode. The design in this work is a potential candidate for mmWave applications that require low-cost components, a low-profile and low DC power consumption, e.g., wireless sensor networks and wireless communication.

## Figures and Tables

**Figure 1 sensors-24-01235-f001:**
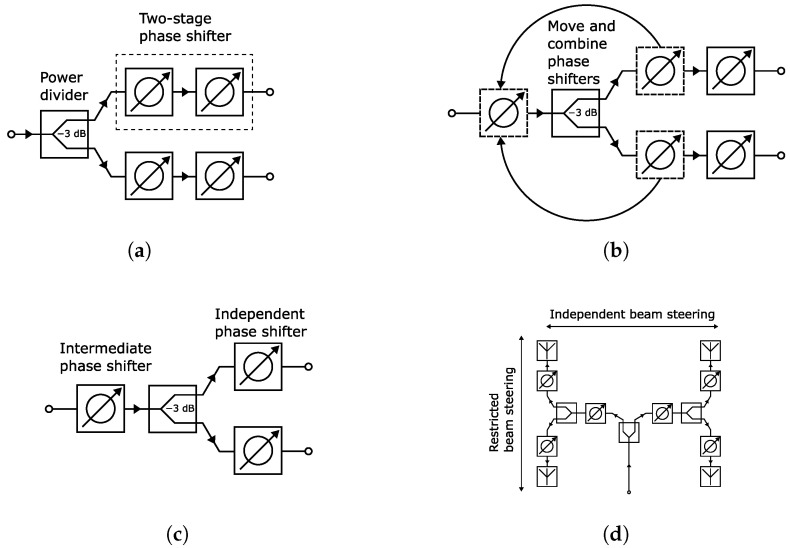
Illustration of the intermediate phase shifting concept. (**a**) The typical implementation of a two-stage phase shifter in a feed network. (**b**) The intermediate step to combine two phase shifters by moving them in front of the power divider. (**c**) The intermediate phase shifting concept. (**d**) Illustration, with a 2 × 2 antenna array, that the horizontal direction has more freedom in beam steering due to phase shift of adjacent elements not being dependent on the intermediate phase shifters.

**Figure 2 sensors-24-01235-f002:**
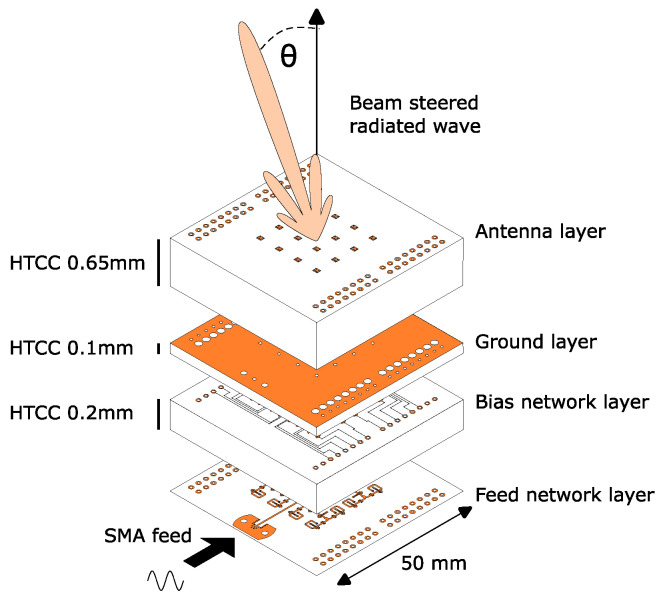
Expanded view of the complete 4 × 4 antenna array. It is fabricated with HTCC and there are four layers in total. The total thickness is 0.95 mm.

**Figure 3 sensors-24-01235-f003:**
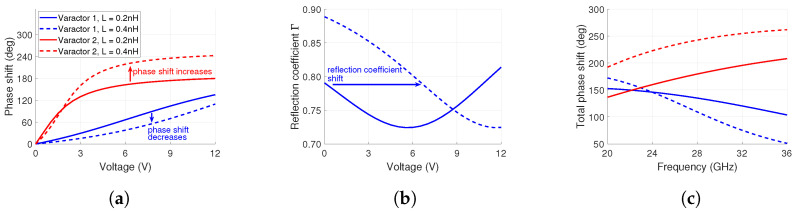
Effect of parasitic inductance on phase shift and reflection coefficient at 28 GHz. (**a**) Phase shift comparison of two varactor models with two inductance values. (**b**) Effect of parasitic inductance on the reflection coefficient of Varactor 1. (**c**) Total phase shift as a function of frequency for the two varactor models when biased from 0 V to 12 V.

**Figure 4 sensors-24-01235-f004:**
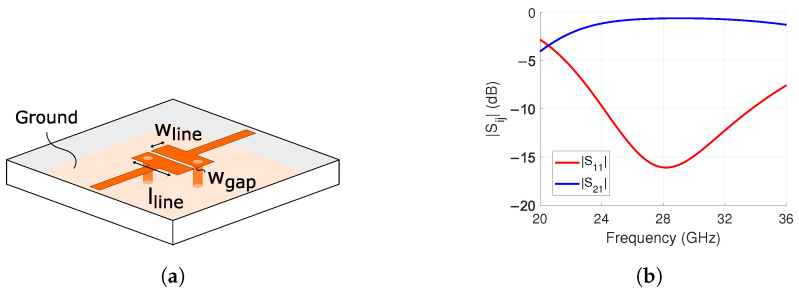
Designed interdigital band-pass filter for connecting L1 and L3 in Figure 5b. (**a**) Structure where the dimensions are $w_{line}$ = 0.26 mm, $w_{gap}$ = 0.1 mm, $l_{line}$ = 1.18 mm. (**b**) Simulation results of $|S_{11}|$ and $|S_{21}|$.

**Figure 5 sensors-24-01235-f005:**
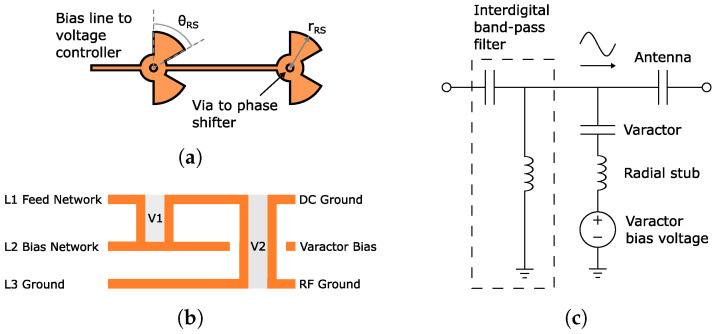
Varactor biasing details. (**a**) Varactor bias line with radial stub which is in layer L2. $\theta_{\mathrm{RS}}=60^{{}^{\circ}}$, $r_{\mathrm{RS}}$ = 0.73 mm, $l_{line}$ = 1.18 mm. (**b**) Feed network layers and their roles in biasing. (**c**) Equivalent circuit of the proposed integrated biasing scheme.

**Figure 6 sensors-24-01235-f006:**
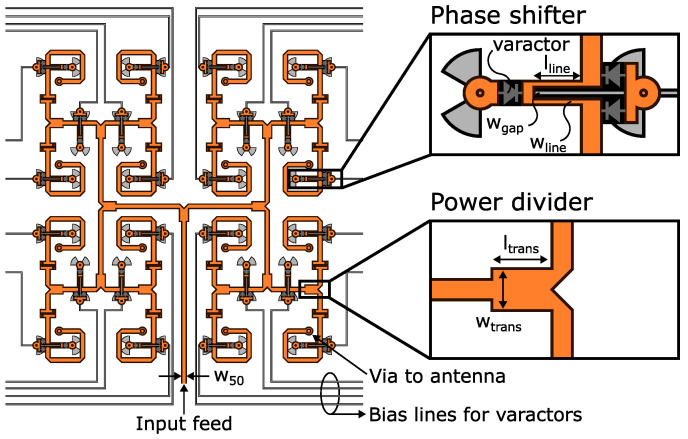
Feed network and its details. The gray radial stubs and lines form the bias network, which is below the transmission lines on layer L2. $l_{\mathrm{line}}=0.8$ mm, $w_{\mathrm{gap}}=0.2$ mm, $w_{\mathrm{line}}=0.1$ mm, $l_{\mathrm{trans}}=1.0$ mm, $w_{\mathrm{trans}}=0.7$ mm, $w_{50}=0.34$ mm.

**Figure 7 sensors-24-01235-f007:**
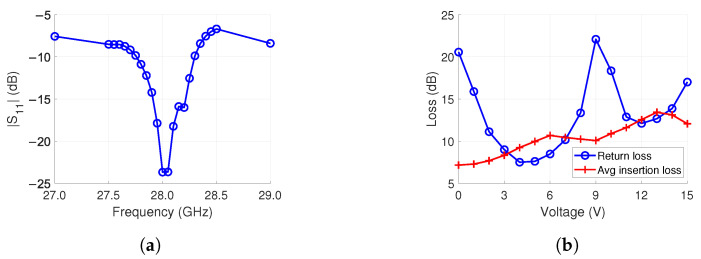
Simulation results of the feed network in Figure 6. (**a**) $|S_{11}|$. (**b**) Return and average insertion loss at 28 GHz when all phase shifters are biased from 0 V to 15 V.

**Figure 8 sensors-24-01235-f008:**
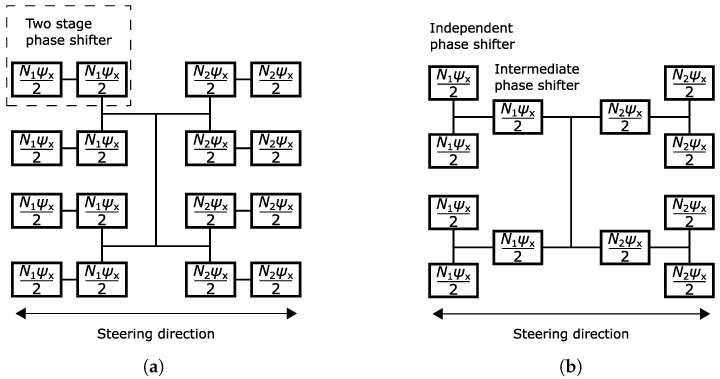
Phase calculation example for a 2 × 4 antenna array containing two-stage phase shifters. (**a**) Conventional antenna array. (**b**) Antenna array with intermediate phase shifting.

**Figure 9 sensors-24-01235-f009:**
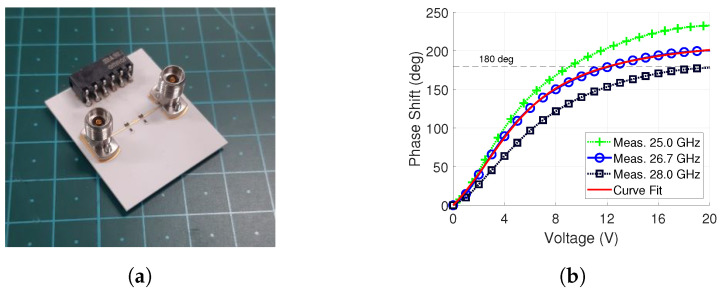
Fabricated phase shifter for determining the phase profile and the measured phase shift along with the curve fit result. (**a**) Phase shifter. (**b**) Phase shift measurement result for one phase shifter and curve fit.

**Figure 10 sensors-24-01235-f010:**
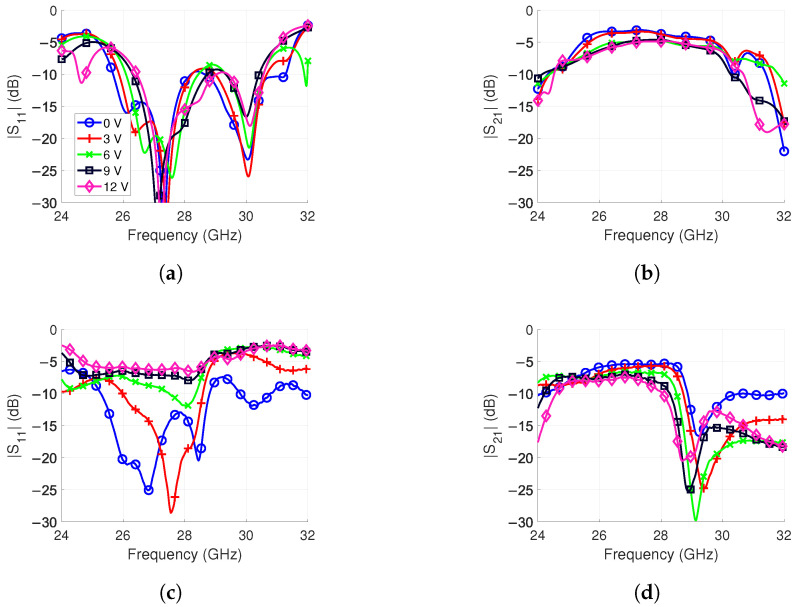
Phase shifter simulation and measurement results when biased from 0 to 12 V. The frequency band is from 24.6 GHz to 28.0 GHz based on the $|S_{21}|$ result. (**a**) $|S_{11}|$ simulation. (**b**) $|S_{21}|$ simulation. (**c**) $|S_{11}|$ measurement. (**d**) $|S_{21}|$ measurement.

**Figure 11 sensors-24-01235-f011:**
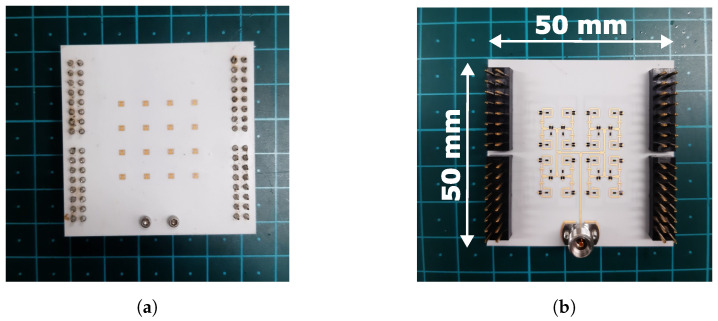
Fabricated antenna array. (**a**) Antenna side. (**b**) Feed network side.

**Figure 12 sensors-24-01235-f012:**
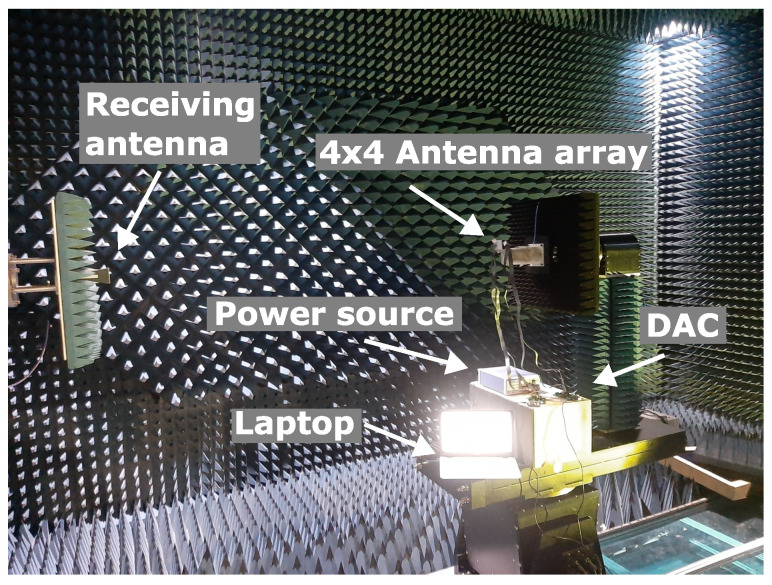
Measurement setup in the chamber.

**Figure 13 sensors-24-01235-f013:**
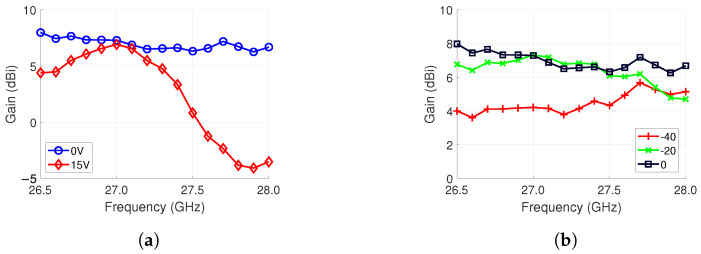
Gain-versus-frequency measurement results. (**a**) All the phase shifters are biased to 0 V and 15 V, respectively. Both cases represent a 0° steering angle, with 0 V being a more optimal case. (**b**) Stable performance at different steering angles is shown.

**Figure 14 sensors-24-01235-f014:**
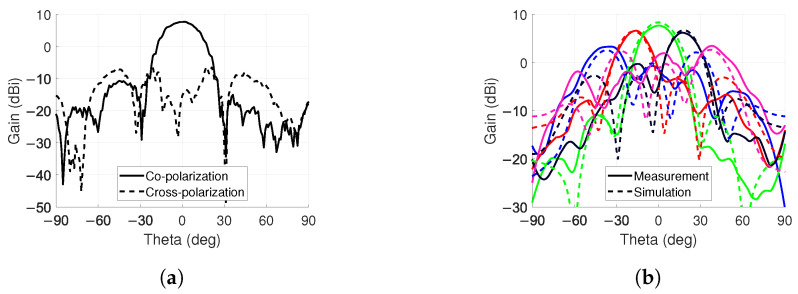
Chamber measurement results. (**a**) Co- and cross-polarization for 0° steering angle on $\phi =90^{{}^{\circ}}$ plane. (**b**) Beam steering measurement and simulation results on $\phi =90^{\circ }$ plane for angles of $\pm 40^{\circ }$, $\pm 20^{\circ }$ and $0^{\circ }$.

**Table 1 sensors-24-01235-t001:** Comparison of maximum steering angles when the progressive phase shift is limited between branches.

Maximum Progressive Phase Shift ($\psi_{\max}$)	Maximum Steering Angle ($\theta_{\max}$)
30°	$\pm 9.6^{{}^{\circ}}$
90°	$\pm 30.0^{{}^{\circ}}$
180°	$\pm 90.0^{{}^{\circ}}$

**Table 2 sensors-24-01235-t002:** Simulated average losses of the feed network from the input to the output at 28 GHz with 0 V bias.

Component	Loss of a Single Component	Number of Components	Total Loss
Power divider	0.2 dB	4	0.8 dB
Interdigital band-pass filter	0.6 dB	1	0.6 dB
Phase shifter	1.1 dB	2	2.2 dB
Conductor	N/A	N/A	2.2 dB
Miscellaneous losses	N/A	N/A	1.2 dB
Feed network average total loss			7.0 dB

**Table 3 sensors-24-01235-t003:** Varactor parameters obtained from curve fit.

Parameter	Value
Junction capacitance $C_{j0}$	0.275 pF
Grading coefficient *m*	1.11
Junction voltage $V_{j}$	0.394 V
Package capacitance $C_{p}$	30.7 fF

**Table 4 sensors-24-01235-t004:** Beam steering results at 26.7 GHz. The control voltages represents the voltage values for each phase shifter in the same column of the antenna array.

Steering Angle (°)	Measured Angle (°)	Control Voltages (V)	Gain (dBi)	SLL (dB)
−50	−42	0, 3.6, 8.6, 2.6	3.1	−2.3
−40	−31	0, 3.1, 6.4, 1.0	4.1	−4.2
−30	−23	0, 2.5, 4.7, 8.3	5.7	−11.1
−20	−18	0, 1.8, 3.2, 4.9	6.9	−9.9
−10	−13	0, 1.0, 1.8, 2.5	7.8	−13.1
0	+2	0, 0, 0, 0	7.7	−18.4
+10	+9	2.5, 1.8, 1.0, 0	7.8	−11.7
+20	+18	4.9, 3.2, 1.8, 0	6.4	−6.1
+30	+23	8.3, 4.7, 2.5, 0	5.1	−4.7
+40	+36	1.0, 6.4, 3.1, 0	3.6	−0.9
+50	+39	2.6, 8.6, 3.6, 0	3.5	−1.6

**Table 5 sensors-24-01235-t005:** Similar recent works in *K*- and *Ka*-band with their main properties.

Reference	Operating Frequency (GHz)	Array Size	Array Type	Steering Method
[13]	24.0 & 26.5	8 × 8	PA ${\;}^{1}$	IC
[16]	28.0	20 × 20	TA SIW ${\;}^{2}$	p-i-n diode
[25]	24.6	6 × 6	TA ${\;}^{3}$	Varactor
[29]	28.0	1 × 10	AA ${\;}^{4}$	p-i-n diode
[30]	28.0	1 × 8	AA	p-i-n diode
[This work]	26.7	4 × 4	AA	Varactor

^1^ Phased array, ^2^ transmitarray w/ SIW feed, ^3^ transmitarray, ^4^ antenna array.

**Table 6 sensors-24-01235-t006:** Similar recent beam-steering works in *K*- and *Ka*-band with performance comparison.

Reference	Steering Range (°) [Plane]	Thickness	Gain (dB) [$\varepsilon_{\mathrm{ap}}^{*}$]	DC Power Consumption	Complete Design
[13]	+41 & +35 [E & H]	0.22$\lambda_{0}$	22.3 [75.1%]	High	Yes
[16]	±40 [E]	3.591$\lambda_{0}$ **	16.2 [3.3%]	Medium	Yes
[25]	±50 [E]	5.990$\lambda_{0}$ **^,+^	−15 [0.03%]	Low	Yes
[29]	−53–+68 [E]	0.093$\lambda_{0}$	11.7 [11.8%]	Medium	No ***
[30]	±50 [E]	0.093$\lambda_{0}$	13.4 [16.4%]	Medium	No ***
[This work]	±40 [E]	0.085$\lambda_{0}$	7.7 [8.8%]	Low	Yes

* Aperture efficiency, ** sum of focal distance and substrate thickness, *** fabricated design is missing reconfigurable components, ^+^ lacking thickness details.

## Data Availability

Available upon request.

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
