# Peer review of "Design of a Compact and Minimalistic Intermediate Phase Shifting Feed Network for Ka-Band Electrical Beam Steering"

_sensors, 2024, doi:10.3390/s24041235_

Round 1

Reviewer 1 Report

Comments and Suggestions for Authors

The paper presents a limited phase-shifter (i.e. less than 360°) for each antenna element to reduce space occupation in a beam-forming network coupled with a 4x4 antenna array working in ka-band.

General comment:

The presented idea is interesting and in line with one of the hot topics in reconfigurable antennas (i.e. reduce complexity and cost in reconfigurable antennas), but it should be clearly state, at the beginning of the paper, the limitation of this reduced phase shifting, e.g.: in this configuration only 1D steering is allowed? 2D steering is allowed with some limitations?

Punctual comments:

Lines 116-118: how much limited is the beam steering?

Lines 125-126: how can you demonstrate the sentence? It could be useful to support with some example (simulation/measurement)

Lines 149-150: how can you support this sentence? PIN diodes are used from years, it seems incredible that no one in the past make this analysis.

Lines 172-175: how did you select those values for C and L? Are in some way related to practical applications, or real values from COTS components?

Lines 209-214: There is no reference to L values, while from line 152, to line 206, you underlined several times the importance and impact of L values on steering. I suggest removing or rephrase Section 3.1.

Lines 220-223: what about 0201 packages?

Lines 246: which SW did you use for this analysis?

Lines 318-321: why you did not perform simulation by including the losses of the metal?

Lines 324: how can the losses (partially) due to the metal roughness since you simulate without including those effects?

Lines 327: I suggest turning the sentence since it is supposed that you vary the voltage and observe the losses.

Lines 402-404: you limit the control voltage to 12V, but from 9V the measured reflection coefficient is not so good (i.e. > -10dB) the impact of antenna performance could be important, and this could be taken into account and underlined.

Table-4: I suggest including aperture efficiency in the comparison, since this is an important parameter to evaluate the overall performance of the reconfigurable antenna.

Comments on the Quality of English Language

As a general comment I suggest to review the paper by reducing some paragraph and the verbosity of some sentence in oder to give a clear view of the concept you want to underline/explain.

Reviewer 2 Report

Comments and Suggestions for Authors

This manuscript proposed a novel phase-shift feeding network for Ka-band beam steering, a compact and minimalistic intermediate phase-shifting feed network. The demonstration of the concept is comprehensive, including both analytical analysis and experimental results. Also, the manuscript is well organized, with step-by-step demonstrations, and is rich in workload and in-depth. A reconfigurable phased array antenna based on varactor diodes is proposed. By moving the phase shift network forward, the size of the feed network is reduced, and the phase of the power-deficient network is reconfigurable through the c-v relationship of the varactor diode, which is the main highlight of the article.

I have several comments for this manuscript.

(1) RF ground may be shown in Figure 4a, and the vertical axis should represent the modulus of the S-parameter in Figure 7 and Figure 10, for example, |S11|.

(2) In Figure 10c, as the control voltage increases in the measurement, the impedance matching of the incident port deteriorates, which may affect the measurement results of beam scanning. It may require a reasonable explanation.

(3) In Figure 11, the size information may be added, or it may be suggested to compare the board size with other items, such as coins.

(4) In Figure 12, a high-resolution image should be shown, and some notes could be added. For example, the array under test, digital to analog converter, or control voltage source.

(5) This phase shift is used for beam scanning using a quadratic active phased array. What is the antenna type for a single element?

(6) Ka-band may have large diffraction. How do the authors avoid or reflect the impact of the dc bias line on radiation? Furthermore, why should the dc layer be placed on top of the feeder layer? Will this not increase the loss and impact?

(7) How do the authors avoid the backward radiation of the antenna array?

(8) In addition to the relationship between dc voltage and phase, what is the relationship between key size parameters of phase shift and phase?

(9) The manuscript reduces the aperture size by reducing the phase shift network, but the aperture efficiency is still not hight. What limits the efficiency?

Comments on the Quality of English Language

The English is OK.

Round 2

Reviewer 1 Report

Comments and Suggestions for Authors

I thanks the authors for providing the review that improve a lot the paper, but there is still some point to be clarified. After this review, I think the paper will be ready for publication.

In the following the comments I would suggest the authors to implement in order to better improve the paper quality and results presentations:

- Line 67-68: the sentence is fine, I kindly suggest to add it also in the abstract, line 13.

- Line 120-128: the paragraph is fine, however, it could be beteer to put some example: I suggest to add a Table for a 2x2 antenna where the phase shift is limited to 30°, 90° and 180° and put the maximum amount of steering you can reach for such kind of values. This will give a clear picture of the achievable performance w.r.t. complexity, compactness, etc.

- Line 140-145: I suggest to review the paragraph, the sentence and the concept inside, seems wrong: if your 2x2 sub-array is controlled by one phase shifter, will become a "macro-element" and the macro-element distance will be much higher thatn lambda/2, then beam steering will be further reduced due to the presence of grating lobes.

- Line 203-207:If you can put some reference and/or applications notes from manufacturer will be appreciated and could be a plus.

Table 3: I suggest adding Voltage values, since Voltage affects the performance of the sub-arrays (due to the high losses variations depending on the voltage).

Table 4: I suggest adding aperture efficiency even for the phased-array [13], oyu can put the comment about the comparison in the text. Moreover, I suggest to review the Table (or to plit it in two Tables) since the number of parameters is really high.

Author Response

Response to Reviewer 1 Comments

Summary: Thank you very much for taking the time to provide additional feedback about this manuscript. The reviewer has provided additional helpful comments and suggestions. Please find the detailed responses below and the corresponding revisions. All changes are highlighted in the re-submitted files as well.

Point-by-point responses to comments:

Point 1:

Line 67-68: the sentence is fine, I kindly suggest to add it also in the abstract, line 13.

Response 1:

The sentence is added to the abstract as well. The abstract has been slightly modified in other ways as well to adhere to the 200 word limit.

Point 2:

Line 120-128: the paragraph is fine, however, it could be beteer to put some example: I suggest to add a Table for a 2x2 antenna where the phase shift is limited to 30°, 90° and 180° and put the maximum amount of steering you can reach for such kind of values. This will give a clear picture of the achievable performance w.r.t. complexity, compactness, etc.

Response 2:

The authors agree with the reviewer and thank for the detailed instructions.

A table has been added (Table 1) to demonstrate the limitations imposed by the limited progressive phase shift between the antennas. Additionally, a paragraph to discuss the contents of the table has been added to Lines 137-151.

Point 3:

Line 140-145: I suggest to review the paragraph, the sentence and the concept inside, seems wrong: if your 2x2 sub-array is controlled by one phase shifter, will become a "macro-element" and the macro-element distance will be much higher thatn lambda/2, then beam steering will be further reduced due to the presence of grating lobes.

Response 3:

The authors thank the reviewer for pointing out this error in the text. This paragraph has been removed and replaced with the discussion provided in the previous point.

Point 4:

Line 203-207:If you can put some reference and/or applications notes from manufacturer will be appreciated and could be a plus.

Response 4:

A reference was added for each inductance value in Lines 209-211.

Point 5:

Table 3: I suggest adding Voltage values, since Voltage affects the performance of the sub-arrays (due to the high losses variations depending on the voltage).

Response 5:

The authors agree with the reviewer.

The voltage values were added to Table 4 (prev. Table 3).

Point 6:

Table 4: I suggest adding aperture efficiency even for the phased-array [13], oyu can put the comment about the comparison in the text. Moreover, I suggest to review the Table (or to plit it in two Tables) since the number of parameters is really high.

Response 6:

The authors agree that Table 4 required a revision. Therefore, the changes described below were made.

The aperture efficiency was added for [13] in Table 6 (prev. Table 4). Additionally, comparison and details of the aperture efficiency for different works were added to Lines 551-556.

Moreover, Table 4 was split into two tables, Table 5 and Table 6. Table 5 contains the main properties of antenna arrays. Operating frequency was added to the table as well to distinguish K- and Ka-band works from each other.

As for Table 6, it contains the most important metrics to compare the different works to each other from previous Table 4. Table formatting was also improved to improve readability.